# Atypical Preeclampsia before 20 Weeks of Gestation—A Systematic Review

**DOI:** 10.3390/ijms24043752

**Published:** 2023-02-13

**Authors:** Jan Modzelewski, Iga Siarkowska, Justyna Pajurek-Dudek, Stepan Feduniw, Katarzyna Muzyka-Placzyńska, Arkadiusz Baran, Anna Kajdy, Magdalena Bednarek-Jędrzejek, Aneta Cymbaluk-Płoska, Ewa Kwiatkowska, Sebastian Kwiatkowski

**Affiliations:** 11st Clinic of Obstetrics and Gynecology, Centre of Postgraduate Medical Education, Żelazna 90 St., 01-004 Warsaw, Poland; 2Department of Obstetrics and Gynecology, Pomeranian Medical University, 70-111 Szczecin, Poland; 3Department of Gynecological Surgery and Gynecological Oncology of Adults and Adolescents, Pomeranian Medical University, 70-111 Szczecin, Poland; 4Department of Nephrology, Transplantology and Internal Medicine, Pomeranian Medical University, 70-111 Szczecin, Poland

**Keywords:** preeclampsia, early onset preeclampsia, atypical preeclampsia

## Abstract

This systematic review was conducted to gather evidence of preeclampsia occurring before the 20th week of gestation, additionally considering the role of PLGF and sFlt-1 in the development of the disease. In the three cases of preeclampsia before the 20th week of gestation presented in the authors’ material, all pregnancies ended up with IUFD, and the SFlt-1/PLGF ratios were significantly elevated in all women. Eligible publications were identified with searches in the PubMed, Embase, Scopus, and Web of Science databases. No date or language restrictions were made. All original peer-reviewed scientific reports were included. A total of 30 publications were included in the final report, including case reports and case series. No other publication types regarding this issue were identified. In the literature, 34 cases of preeclampsia with onset occurring before the 20th week of gestation were identified, for a final total of 37 cases. Live births were reported in 5 cases (10.52%), and there were 9 intrauterine fetal demises (24.32%), and 23 terminations of pregnancy (62.16%). Preeclampsia before the 20th week of gestation is rare but can occur. We collected all available evidence regarding this phenomenon, with 37 cases reported worldwide. We call for large-scale cohort or register-based studies to establish revised definitions or develop new ones regarding the currently unrecognized very early onset preeclampsia.

## 1. Introduction

Preeclampsia (PE) is a known obstetric pathology, the definition of which, due to the ongoing process of understanding its pathogenesis, is still evolving. Originally, it was known as a condition of hypertension and proteinuria, but then the definition was expanded to include liver and kidney dysfunctions, thrombocytopenia, and fetal growth restriction. In 2018, the International Society of the Study of Hypertension in Pregnancy published a definition of preeclampsia. The diagnostic criteria include newly onset hypertension ≥20 weeks of gestation, additional proteinuria, and maternal organ dysfunction or uteroplacental dysfunction [1].

The process of trophoblast invasion begins at the early stages of pregnancy. The most intense transformations take place before the 18th week of gestation. In the primary period, a structure called a cytotrophoblast shell is created, the task of which is to protect the embryo from the harmful effects of oxygen and provide nourishment without the contribution of all the mother’s blood [2,3].

After the dilatation of the spiral arteries after about 12 weeks of pregnancy, the mother’s blood flows into the uteroplacental space around the chorionic villi for the first time. A further transformation of the spiral vessels occurs, which extends from the central parts of the placenta to its peripheral parts. Moreover, the surfaces of the villi keep enlarging, thanks to which the exchange capacity of the uteroplacental compartment is constantly increasing. This process is inextricably linked to the increasing demands of the fetus [2,4,5].

Nowadays, the diagnosis of early preeclampsia is associated with the simultaneous occurrence of increased resistance in the blood flow of uterine arteries and disturbances in the angiogenesis process, which, to some extent, can be monitored using angiogenesis placental markers, such as soluble FMS-like tyrosine kinase-1 (sFlt-1) and placental growth factor (PLGF). SFlt-1 and PLGF demonstrate altered values 6–8 weeks before clinical diagnosis [6].

Still, a criterion for the diagnosis of preeclampsia is its occurrence after the 20th week of gestation. With a better understanding of what causes this pathology, we decided that we should question this criterion and assess if a new definition of preeclampsia should be elaborated. The aim of this systematic review was to collect and assess all data regarding preeclampsia with onset before the 20th week of gestation.

It has been accepted to define preeclampsia as a condition developing after the 20th week of pregnancy. Do the symptoms of placental insufficiency really only occur in the second half of pregnancy? If the clinical picture completely coincides with preeclampsia in the first half of pregnancy, with reasonable suspicion or certainty of chorionic villi damage and features of placental insufficiency, should we not recognize that we are dealing with preeclampsia? This review aimed to investigate whether we should again modify its definition.

### PLGF and sFlt-1

Placental development is a sequence of events. It starts soon after the blastocyst implantation in the maternal decidua. Sysytiotrofoblast forms the lacune, which is later filled with maternal blood. Simultaneously, extravillous trophoblast starts to invade the maternal spiral arteries, forming plugs that stop the maternal blood flow and replace the maternal muscular cells in the zona intima of the vessels and, temporally, the maternal endothelium. Arterial remodeling is controlled vastly by the immune system, namely uNK and Treg cells. What is interesting is that uNK cells have the ability to secrete angiogenic factors, including PLGF, which plays role in uNK proliferation [7]. Higher PLGF expression in uNK cells was confirmed in high-risk pregnancies by high-resistance flow in the uterine arteries [8]. A high expression of PLGF is observed in arrested implantation sites. One of the explanations for the dual role of PLGF in the development of abnormal vasculature and failed implantation may be an insufficient expression of VEGF in those cases, with the role of PLGF switching from angiogenesis to inflammation [9].

Altered arterial remodeling or arterial plug resolution leads to oxidative stress of the syncytiotrophoblasts, resulting in decreased PLGF production. Degenerative syncytiotrophoblastic cells, or syncytial knots, are known to produce large amounts of sFlt-1 [7]. The fates of 237 patients with extremely high PLGF/sFlt-1 ratio (>655) have been investigated. At inclusion, 185 of them had preeclampsia (78.1%), and the remaining mostly suffered from FGR (82.7% of the rest). The maternal complications observed were severe hypertension (19.5% with PE, 5.8% without PE), oliguria (8.1% with PE, 0% without PE), and placental abruption (8.7% with PE, 21.2% without PE) [10].

PLGF is a member of the vascular endothelial growth factor (VEGF) family. PLGF is secreted as a homodimer but can also form a heterodimer PLGF/VEGF. Members of the VEGF family are potent vascular growth factors [11].

PLGF is coded by the PGF gene, located on chromosome 14q24.3 [12]. Currently, there are four known isoforms of the PLGF: PLGF-1 contains 149 amino acids, PLGF-2 contains an additional heparin binding sequence with 21 amino acids, PLGF-3 has an in-frame insertion loop with 72amino acids, PLGF-4 has both the additional 72-amino-acid sequence of PLGF-3 and the 21-amino-acid heparin binding sequence. PLGF-1 and PLGF-3 bind primarily to VEGFR1 (Flt-1), while the PLGF-2 main receptor is the neuropilin-1 receptor (NRP). PLGF-4 may be a cell membrane-associated variant, with autocrine abilities. Isoforms-1, -2, and -3 have confirmed the ability to bind to sFlt-1 [12,13].

PLGF levels are undetectable in most healthy tissues. On the other hand, they have a role in angiogenesis in pathological conditions, such as inflammation or ischemia. Under typical conditions, PLGF expression is upregulated by hypoxia, inflammatory cytokine growth factors, and hormones—all those present during implantation and placenta development. However, hypoxia during placentation downregulates PLGF expression and upregulates sFlt-1 expression. What is more, PLGF expression in the placenta is independent of the activity of the hypoxia-induced factor (HIF)-1. The trophoblast is the main source of PLGF. Its production in the trophoblast is strongly upregulated in early pregnancy [14].

The PLGF homodimer binds to the FLT-1 (VEGFR-1). FLT-1 suppresses angiogenic signals and modulates the behavior of immune cells. It increases the proliferation, survival, and migration of the macrophages. It also promotes the expressions of cytokines, assuming a role in decidualization [14].

FLT-1 (VEGFR-1) has a higher affinity to VEGF and PLGF but lower kinase activity than VEGFR-2. VERFR-2 has lower affinity but much higher activity. VEGFR-1, therefore, plays a dual role as a weak promoter of angiogenesis via its kinase activity while depleting the pool of available VEGF for VEGFR-2 and, therefore, having an antiangiogenic effect [15]. PLGF has a binding potential to VEGFR-1 but not VEGFR-2, displacing VEGF from VEGFR-1 and increasing the availability of VEGF to VEGFR-2 [9,16].

FLT-1 is a receptor for both VEGF and PLGF. It is coded by the FLT gene, a member of the src family, located on chromosome 13q12.3. The gene name comes from its close similarity to the FMS gene. Flt-1 possesses tyrosine kinase activity, controlling cell proliferation and differentiation. The soluble form of Flt-1 (sFlt-1) is a product of alternative splicing. SFlt-1 binds to PLGF and VEGF with high affinity, decreasing its bioavailability [17].

FLT-1 co-receptor neuropilin (NRP)-1 and NRP-2 are expressed in axons, vessels, and immune cells. NRP-1 is expressed mostly in dendritic cells (DCs) and T regulatory (Treg) cells. It is constitutively expressed by CD4^+^CD25^high^ natural Treg cells. It promotes prolonged interaction between Tregs and immature DCs. This interaction probably slows DC maturation and therefore promotes immune tolerance. Conversely to the production of a soluble form of FLT-1 receptor—a soluble form of the NRP-1—the sNRP-1 is detected. SNRP-1 binds to both PLGL and VEGF, also decreasing its bioavailability [14].

Activation of the FLT-1 receptor has also a role in the immune response via transcription factor NF-κB. NF-κB has binding sites in the promoter region of PLGF and can modulate its expression. NF-κB is involved in immune response, angiogenesis, and hypoxia processes present at the formation of the placenta [14].

High levels of sFLT-1 are observed in post-partum cardiomyopathy and individuals with an altered myocardial performance index. Patten et al. suggested that a high level of sFlt-1 together with poor local pro-angiogenic defense in one heart may lead to the development of the disease [18]. An altered expression of Flt-1 may be one of the factors leading to fetal demise in malaria infection [19].

SFlt-1 has multiple splice variants, but two are predominant. SFlt-1-1 (sFlt1_v1, sFlt-1-i13) is expressed in several tissues, while sFlt-1-14 (sFlt-1-v2, sFlt-1 -e15a) is expressed exclusively in a placenta. The placental expression of the sFlt-1_1 is roughly three times larger than the sFlt-1-14 splice variant. In a preeclampsia expression of both variants is increased, but the sFlt-1-1 to a larger extent [20,21].

SFlt-1 expression is regulated by the renin–angiotensin system in the placenta, and angiotensin II is a stimulator of sFlt-1. Ex vivo studies have shown the potential role of the proton pump inhibitors (esomeprazole) in mitigating this effect. In ex vivo conditions, esomeprazole decreased the expression of the sFlt-1 mRNA and the concentration of sFlt-1 in the supernatant of a healthy and a preeclamptic placenta. The probable mechanism of action may be the inhibition of the uptake of angiotensinogen or (pro)renin transport [22]. Data on the use of proton pump inhibitors in preeclamptic women show varying results [23,24,25,26].

The treatment of hypertension with alphamethyldopa had different effects on PLGF and sFlt-1. The effect also varied between preeclamptic and gestational hypertension patients. Alphamethyldopa treatment did not change the levels of either factor in the gestational hypertension patients. In the preeclamptic patients, on the other hand, the levels of sFlt-1 decreased while the PLGF remained stable. The effect was visible both in the placenta and the serum, as changes in the serum were mirrored by changes in the placenta. It is speculated that this is the effect of the alphamethyldopa stimulation of the α2β-adrenoceptors on the placenta and the myometrium, and thus, directly influences sFlt-1 production [27]. On the other hand, data regarding the mechanism of action is scarce, and this effect has not been confirmed by an ex vivo study [28].

The association between the levels of the PLGF and sFlt-1 and placenta accreta spectrum (PAS) was recently summed up in a meta-analysis. The authors concluded that the PLGF levels did not change in PAS, but the results of the studies were conflicting. The sFlt-1 was significantly decreased and not dependent on the type of PAS, and these results were consistent among studies. A suitable cut-off value has to be established to include these promising findings in clinical practice [29].

Female and male placentas have different expressions of angiogenic factors. The sex-dependent expression of a plethora of factors was found as early as the blastocyst stage. The same was confirmed for the sFlt-1 and the PLGF in the first trimester. The mothers of female fetuses had higher PLGF and sFlt-1 levels in the first trimester. This finding is concordant with the known richer vasculature of female placentas. As a result of such vasculature, female placentas have greater placental capacity [30].

## 2. Case Presentation

We present three cases of preeclampsia with onset before 20 weeks of gestation from our centers. Cases 1 and 2 were patients treated at the Pomeranian Medical University Hospital, in northwest Poland. The third case came from the “Zelazna” Medical Centre in Warsaw, Poland. Both hospitals are tertiary reference centers. The only inclusion criterium was the onset of the symptoms of preeclampsia before the 20th week of gestation, with no exclusion criteria (Table 1). According to Polish law, the presentation of anonymous retrospective patient data does not need bioethical committee supervision.

The results for the PE onset, comorbidities, and outcomes are presented in summary form, the presented cases, and a systematic review.

Of note, the angiogenesis markers and the uterine artery flow were markedly altered in all three cases. One woman had a high preeclampsia risk estimation, according to Fetal Medicine Foundation protocol, at the first-trimester examination (1:4).

## 3. Materials and Methods

The subject-related articles were processed according to Preferred Reporting Items in Systematic Reviews and Meta-Analyses (PRISMA) guidelines.

All original scientific articles regarding the cases of atypical preeclampsia before the 20th week of gestation were assessed. We excluded letters to the editor, opinions, and non-peer-reviewed articles. All cases of preeclampsia that occurred after the 20th week of gestation were excluded.

The inclusion criteria were, similar to the classic definition, newly onset hypertension or worsening of chronic hypertension, and additional proteinuria, maternal organ dysfunction, or uteroplacental dysfunction.

On the 13 June 2021, we searched PubMed, Embase, Scopus, and Web of Science databases, using the search phrase. Search engine-specific options limiting the search to the title and abstract were used. No date or language restrictions were used. The articles we found that were eligible for this search were written in English and Spanish (languages of the authors ).

Each report was assessed independently by two authors. The PRISMA chart of the whole process is shown in Figure 1.

Two authors (IS and JPD) independently extracted data regarding atypical preeclampsia before 20 weeks of gestation. The data that were extracted from the full texts included the onset week of gestation when preeclampsia occurred, gravidity, parity, type of pregnancy (single, twin, partial molar, complete molar, or IVF), comorbidities, information on first-trimester prenatal screening, the evolution of blood pressure values, proteinuria, markers of angiogenesis, information on the uterine artery PI, and the pregnancy outcome. No missing information was assumed. All articles included in the review are case reports, and therefore, there was no risk of bias assessed.

## 4. Results

There were 30 papers describing 34 cases of atypical preeclampsia found in the reviewed literature that met the criteria for this systematic review. The earliest reported case of preeclampsia was at 14 weeks of gestation. The data on the onset of preeclampsia and the end of pregnancy are shown in Figure 2. There was no information on first-trimester preeclampsia screening. Most of the patients were admitted to the hospital with high blood pressure and proteinuria. Information about the post-pregnancy events was available for 12 cases—the blood pressure was normalized after pregnancy in all women who were not diagnosed with hypertension before the pregnancy. The angiogenesis markers reported in the three studies are shown in Table 2. In total, there were 23 terminations of pregnancy, 9 cases of intrauterine fetal demise, and 5 cases of preterm live deliveries, including two selective terminations of one fetus with the live birth of another twin/triplet. Two live births were cesarean sections. The majority of women had pre-pregnancy morbidities, predisposing them to the development of preeclampsia during pregnancy. The data on the outcome and maternal comorbidities are shown in Figure 3. A summary of the included studies is presented in Table 3. Rejected papers and reasons for rejection are presented in Table 4.

## 5. Discussion

The International Society of the Study of Hypertension in Pregnancy’s definition of preeclampsia excludes the possibility of preeclampsia before 20 weeks of gestation. In our work, we indicate that in all analyzed cases, preeclampsia occurred before the 20th week of pregnancy. Furthermore, we show that women can develop preeclampsia in normal and molar pregnancies.

The majority of the women who develop atypical PE had pre-existing co-morbidities, mainly anti-phospholipid syndrome, pre-pregnancy hypertension, and protein S deficiency. Infertility treatment may be a risk factor for atypical PE. This may be a factor worth considering in the development of the definition of atypical preeclampsia.

Five of the presented cases resulted in the delivery of a live-born preterm child. One may consider atypical preeclampsia as extremely severe; however, there is a chance of fetal survival.

In our presented cases, the women diagnosed with very early onset preeclampsia had severe angiogenesis imbalance, detectable by the sFlt-1/PLGF ratio. Data regarding the first-trimester preeclampsia screening were available for only one woman, with a risk of 1:4, but it would be reasonable to expect similarly high results for all women endangered with very early onset preeclampsia. The answer to the question of the effectiveness of acetylsalicylic acid prophylaxis in these cases is extremely interesting but beyond the scope of this review.

In light of the available evidence, it is not possible to assess the performance of first-trimester screening in the prediction of atypical preeclampsia. On the other hand, all our presented cases had high a highly altered PLGF/sFlt-1 index at the onset of the disease. PLGF is recommended in preeclampsia screening according to the Fetal Medicine Foundation, and therefore, we think that screening including all recommended maternal characteristics, blood pressure, uterine arteries flow, and biochemical markers, including the PLGF, may be enough to screen for atypical preeclampsia.

It should be noted that our review includes case reports only. Therefore, the assessment of study bias was impossible. We also cannot exclude publication bias. Moreover, presented cases do not fit the common definition, and therefore, it may be that some cases have not been reported due to misdiagnosis. The results of our study are affected by these limitations, but in our opinion, they rather underestimate the occurrence of atypical preeclampsia.

We performed an extensive search in four databases with very few limitations. The identified studies were inconsistent with the reported data. The extraction of some predefined data was therefore impossible, including information on first-trimester perinatal screening, markers of angiogenesis, and measures of uterine artery perfusion. To our knowledge, this is the first systematic review regarding this topic.

Based on our review, we cannot determine new criteria for the preeclampsia definition before the 20th week of pregnancy. The reviewed studies lack uterine artery Doppler flow, angiogenesis placental markers, and first-trimester preeclampsia screening data, which may become a part of the definition. A comprehensive cohort or a large-scale database study should be conducted to deliver the data required to extend the preeclampsia definition to cases before the 20th week of pregnancy.

## 6. Conclusions

Preeclampsia before the 20th week of gestation is rare but it can occur. We collected all available evidence regarding this phenomenon, with 37 cases reported worldwide. We call for large-scale cohort or register-based studies to establish revised definitions or develop new ones regarding the currently unrecognized very early onset preeclampsia.

## Figures and Tables

**Figure 1 ijms-24-03752-f001:**
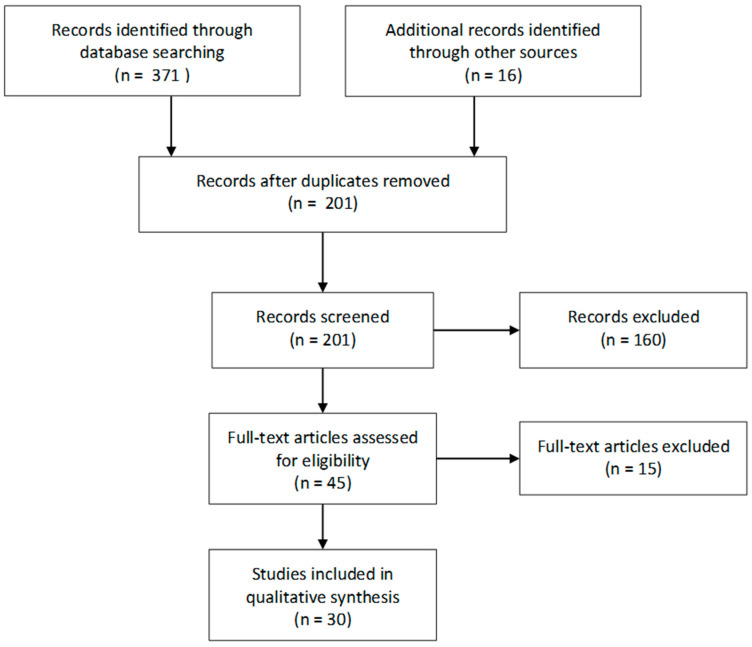
PRISMA flow chart.

**Figure 2 ijms-24-03752-f002:**
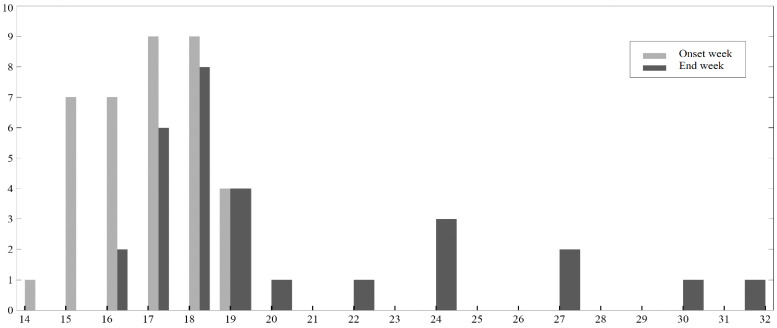
Preeclampsia onset week and the week of the pregnancy end by the number of cases reported.

**Figure 3 ijms-24-03752-f003:**
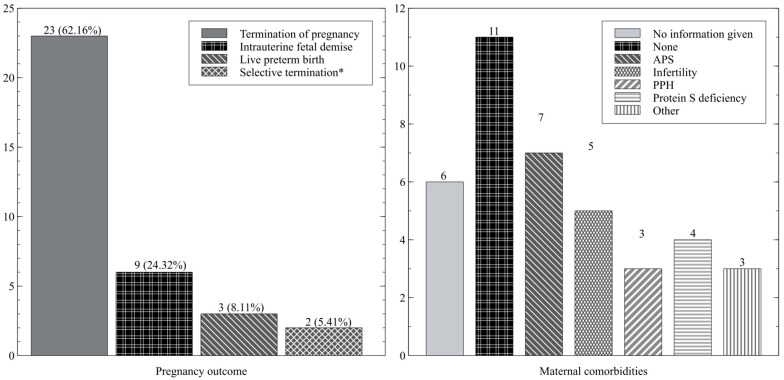
Pregnancy outcome—number of cases (%); maternal comorbidities—number of cases. * Selective termination and live birth of other twin/triplets. APS—antiphospholipid syndrome, PPH—pre-pregnancy hypertension.

**Table 1 ijms-24-03752-t001:** Presentation of cases. APS—antiphospholipid syndrome, FMF—Fetal Medicine Foundation, IUFD—intrauterine fetal demise, UtA—uterine artery, PI—pulsatility index, aCGH—array comparative genomic hybridization.

	Case 1 (G1P0)	Case 2 (G4P1)	Case 3 (G1P0)
Onset week	16	15	17
Comorbidities	APS, history of thrombosis	DM, obesity	Idiopathic thrombocytopenia
First-trimester screening—risks, according to FMF protocol	Trisomy—low; no PE screening, no fetal malformations	Trisomy—low; PE—1:4, PLGF—13.3 pg/ml (0.33 MoM)	Trisomy 21 1:340, NIPT—low risk; no PE screening, PAPP-A 0.7 MoM
Angiogenesis markers at the onset	Sflt-1—8956 pg/mL; PLGF—9.96 pg/mL; sFlt-1/PLGF ratio—899	sFlt-1—3540 pg/mL; PlGF—9.21 pg/mL; sFlt-1/PlGF ratio—393	PLGF—20.5 pg/mL; sFlt—10610.0 pg/mL; sFlt-1/PLGF ratio—517.6
Uterine artery flow at the onset	Left UtA PI—3.04; right UtA PI—3.55; mean PI—3.295 > 95th percentile	Left UtA PI—3.7; right UtA PI—2.89; mean PI—3.295 > 95th percentile	Left UtA PI—2.25; right UtA PI—2.1; mean PI—2.175 > 95th percentile
Outcome	IUFD at 19 weeks, placental aCGH normal	IUFD at 15 weeks, placental aCGH normal	IUFD at 24 weeks, anhydramniosis, birth of a stillborn female fetus 370 g

**Table 2 ijms-24-03752-t002:** Reported angiogenesis markers. Soluble FMS-like tyrosine kinase-1 (sFlt-1)/Placental growth factor (PLGF).

Case	Yoneda et al. [31]	Romero-Arauz et al. [32]	Suzuki et al. [33]
Reported angiogenesis markers	At onset: sFlt-1—4196 pg/mL; PLGF—127 ng/mL, sFlt-1/PLGF ratio—476	At onset: sFLT-1/PLGF ratio—895.5	At 21 weeks (onset at 16): sFlt-1—13,400 pg/mL; PlGF, —21.9 pg/mL; sFlt-1/PlGF ratio—611.9

**Table 3 ijms-24-03752-t003:** Presentation and summary of included cases. Papers (n = 30) and cases (n = 34) were included in the systematic review. HBD—hebdomas, weeks of pregnancy when pregnancy ended, IUFD—intrauterine fetal demise, LB—live birth, sFGR—selective fetal growth restriction, RDS—respiratory distress syndrome, TOP—termination of pregnancy.

Case	Authors:	Onset Week	Pregnancy	Comorbidities	Outcome
1	Imasawa et al. [31]	15	G1P0, dichorionic twin	12 cm leiomyoma	TOP 18 HBD
2	Suzuki et al. [32]	16	G5P1, singleton	Protein S deficiency, infertility	Cesarean section at 24 + 3 HBD, LB 303 g, retinopathy, RDS, rickets
3	Stefos et al. [33]	18	G1P0, partial molar, growth-restricted fetus	none	TOP 18 HBD
4	Yoneda et al. [34]	19	G1P0, partial molar, growth-restricted fetus	none	TOP 20 HBD
5	Myer et al. [35]	15	G3P2, twin surrogate IVF pregnancy	Chronic hypertension, asthma, positive ANA-antibodies	IUFD 9 and 16 HBD
6	Hazra et al. [36]	18	G4P3, singleton	Thyroidectomy at 9-year-old	TOP 18 HBD
7	Bornstein et al. [37]	15	G4P0, singleton	Obesity	TOP 17 HBD
8	Alsulyman et al. [38]	17	G2P1, singleton, admitted because of upper abdominal pain and biochemical markers of HELLP	Antiphospholipid syndrome	IUFD 17 HBD
9	Alsulyman et al. [38]	19	G3P2, singleton	Antiphospholipid syndrome	IUFD 19 HBD
10	Alsulyman et al. [38]	17	G3P2, singleton	Antiphospholipid syndrome	IUFD 17 HBD
11	Stillman et al. [39]	15	G1P0, fetal reduction to twins, IVF	PCOS	TOP, HBD not given
12	Parrott et al. [40]	18	G3P0, singleton	None	TOP 18 HBD
13	Tanaka et al. [41]	17	G1P0, singleton, admission because of isolated leg edema	None	IUFD 22 HBD
14	Maya et al. [42]	19	G1P0, singleton	None	TOP, HBD not given
15	Mayer-Picke et al. [43]	17	G2P1, singleton, admitted because of abdominal pain; plasma exchange after admission	Antiphospholipid syndrome	LB 27 HBD, normal development
16	Mayer-Picke et al. [43]	17	G1P0, singleton, admitted because of abdominal pain; plasma exchange after admission	Antiphospholipid syndrome	TOP 24 HBD
17	Mayer-Picke et al. [43]	19	G1P0, singleton, admitted because of thrombocytopenia; plasma exchange after admission	Antiphospholipid syndrome	LB 27 HBD, normal development
18	Rodríguez et al. [44]	16	G1P0, singleton, partial molar	None	TOP 16 HBD
19	Romero-Arauz et al. [45]	18	G1P0, singleton, IVF	Chronic hypertension, infertility	TOP 18 HBD
20	Konstantopoulos et al. [46]	18	G1P0, twin after double donated embryo transfer, sFGR of one fetus	Infertility	Selective reduction of sFGR twin, CS 30 HBD, retinopathy of prematurity
21	Khan et al. [47]—no full text available, data from abstract	15	G2P0, singleton, IVF	Infertility	TOP 17 HBD
22	Thomas et al. [48]	14	G6P4, singleton	Chronic hypertension, obesity, cholecystectomy	TOP 22 HBD
23	Craig et al. [49]	17	G1P0, singleton, kariotype 69 XXY	None	TOP 17 HBD
24	Nwosu et al. [50]	18	G2P1, dichornionic twin	None	TOP, HBD not given
25	Billieux et al. [51]	18	G2P1, partial molar	None	TOP 18 HBD
26	Brittain and Bayliss [52]	18	G7P2, partial molar	No information given	TOP, HBD not given
27	Es Saad et al. [53]	16	G2P1, complete molar	None	TOP, HBD not given
28	Rahimpanah and Smoleniec [54]	16	G2P1, partial molar	No information given	TOP, HBD not given
29	Nugent et al. [55]—no full text available, data from abstract	15	Twin pregnancy, one fetus normal, another partial molar	No information given	TOP of the partial molar fetus 15 HBD, normal twin 19 HBD
30	Prasannan-Nair et al. [56]	17	G1P0, partial molar	No information given	TOP at 19 HBD
31	Sherer et al. [57]	17	G2P1, partial molar, fetal karyotype 69, XXY	No information given	TOP at 17 HBD
32	Haram et al. [58]	18	G4P0, singleton, admitted due to epigastric pain	Antiphospholipid syndrome, protein S deficiency	TOP at 18 HBD
33	McMahon et al. [59]	16	G1P0, singleton, admitted due to epigastric pain	None	IUFD at 18 HBD
34	de Weg et al. [60]—no full text available, data from abstract	16	Triplet pregnancy	No information given	Selective reduction of one monochorionic twin, delivery in 32 HBD

**Table 4 ijms-24-03752-t004:** Rejected after full-text review.

	Author	Title	Reason for Rejection
1	Stevens et al. [61]	Atypical preeclampsia—gestational proteinuria	PE > 20 weeks of gestation
2	Albayrak et al. [62]	Atypical preeclampsia and eclampsia: report of four cases and review of the literature	PE > 20 weeks of gestation
3	Castelazo-Morales et al. [63]	Atypical preeclampsia and perinatal success: a case report	PE > 20 weeks of gestation
4	Ditisheim et al. [64]	Atypical presentation of preeclampsia	No text available
5	Rojas-Arias et al. [65]	Characterization of atypical preeclampsia	PE > 20 weeks of gestation
6	Valle Tejero et al. [66]	Classic vs atypical HELLP syndrome. Obstetric and perinatal results	No text available
7	Sibai and Stella [67]	Diagnosis and management of atypical preeclampsia-eclampsia	No data on PE < 20 weeks presented
8	Van Scheltinga et al. [68]	Differentiating between gestational and chronic hypertension; an explorative study	No data on the timing of PE onset
9	Van Scheltinga et al. [69]	Hypertension before 20 weeks’ gestation and chronic hypertension	No text available
10	Seguro et al. [70]	Management of arterial hypertension before 20 weeks’ gestation in pregnant women	No data on PE < 20 weeks presented
11	Zhang et al. [71]	Plasma level of placenta-derived macrophage-stimulating protein-chain in preeclampsia before 20 weeks of pregnancy	No data on PE < 20 weeks presented
12	Sáez Cantero et al. [72]	Preeclampsia and eclampsia with atypical presentation	No data on PE < 20 weeks presented
13	Stella and Sibai [73]	Preeclampsia: Diagnosis and management of the atypical presentation	No data on PE < 20 weeks presented
14	Sibai et al. [74]	Eclampsia in the first half of pregnancy. A report of three cases and a review of the literature	No text available
15	Garland et al. [75]	TMA in pregnancy before 20 weeks’ gestation: Is this preeclampsia or primary TMA?	No data on PE < 20 weeks presented

## Data Availability

Not applicable.

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
