# Peer review of "Atypical Preeclampsia before 20 Weeks of Gestation—A Systematic Review"

_ijms, 2023, doi:10.3390/ijms24043752_

Round 1

Reviewer 1 Report

It s an interesting systematic review on a controversial subject of very early-onset PE. In my opinion it needs some changes before possible publication

- I would not write on Your own case reports in a systematic review , I would mentioned about it in Discussion ( the table You presented on these cases may be added later in the text after You mentioned about it in Discussion) or You will describe 3 clinical cases but not in the study like this ( Systematic Review)

- You have to include precisely the inclusion criteria of early onset PE diagnosed in the studies You included into Systematic Review. Do remember that increased hypertension is a essential basic criteria, not the sFlt-1/ PlGF ratio. 

- In the Discussion section may be it worth mentioning that may be some primary maternal disease predispose to early - onset, atypical PE ; especially APS, some connective tissue diseases, may be in these cases PE could be diagnosed before 20 weeks 

Author Response

Reviewer 1:

Reviewer advice:

Answer:

I would not write on Your own case reports in a systematic review , I would mentioned about it in Discussion ( the table You presented on these cases may be added later in the text after You mentioned about it in Discussion) or You will describe 3 clinical cases but not in the study like this ( Systematic Review)

It's not common to present own cases and include them in a systematic review, but on the other hand, it may be permissible, especially when authors intend to show the bigger picture. We include examples below.

https://www.ajog.org/article/S0002-9378(22)00316-7/fulltext

https://www.thelancet.com/journals/laninf/article/PIIS1473-3099(19)30041-6/fulltext

You have to include precisely the inclusion criteria of early onset PE diagnosed in the studies You included into Systematic Review. Do remember that increased hypertension is a essential basic criteria, not the sFlt-1/ PlGF ratio. 

Thank you for this comment, in fact, we did not describe inclusion criteria. Corrected in verses 219-221:

The inclusion criteria were, likewise to the classic definition, newly onset hypertension or worsening of chronic hypertension, and additional proteinuria, maternal organ dysfunction, or uteroplacental dysfunction.

- In the Discussion section may be it worth mentioning that may be some primary maternal disease predispose to early - onset, atypical PE ; especially APS, some connective tissue diseases, may be in these cases PE could be diagnosed before 20 weeks

We agree. The additional description made in discussion (293-296):

“Majority of the women who develop atypical PE had pre-existing co-morbidities mainly anti-phospholipid syndrome and pre-pregnancy hypertension and protein S deficiency. Infertility treatment may be a risk factor for atypical PE. This may be a factor worth considering in the development of the definition of atypical pre-eclampsia.”

Reviewer 2 Report

Dear Authors,

Opinions about the pathogenesis of preeclampsia are still the subject of active discussion between scientists and clinicians. And this common pregnancy complication is seen as a combination of insufficient trophoblast invasion, placental hypoxia, pro-inflammatory immune environment, anti-angiogenic factors, endothelial dysfunction, and oxidative stress. However, data obtained over the past 20 years have made some adjustments regarding preeclampsia as a condition that is caused by inherent maternal cardiovascular dysfunction, perhaps entirely independent of the placenta. Various phenotypes of preeclampsia have been described that are associated with opposite manifestations in terms of cardiac output, vascular resistance, and intravascular volume. In this regard, in my opinion, atypical preeclampsia, manifesting up to 20 weeks, can probably be considered as a latent form.

It should be noted that the authors, after conducting an extensive literature search, presented the quintessence of cases that are classified as atypical preeclampsia, supplementing them with their own observations. However, in general, I have the impression that the manuscript is presented as "Original Research". And a number of comments that I have, give reason to recommend that this manuscript be included in the “Research Article” section and provide a more detailed description of their own observations in the context of data from other authors.

Remarks are given below.

Abstract. This section should not include titles (1)Introduction, (2)Materials and methods: , (3)Results:. In addition, I did not see an analysis of evidence regarding the role of PLGF and sFlt-1 in the development of the disease in the manuscript, since the authors themselves indicated the absence of data on the levels of placental factors in the presented search cases.

The authors also indicated "No risk of bias was accessed." However, in Discussion is indicated " We also cannot exclude publication bias." I do not quite understand what data this phrase refers to.

In the phrase «Three cases of preeclampsia before the 20th week of gestation. All pregnancies ended up with IUFD. SFlt-1/PLGF ratios were significantly 22elevated in all women.» it is necessary to indicate that these cases are related to the authors data, otherwise it is not clear.

Introduction. This section does not fully explain the pathophysiology of preeclampsia, since by now a sufficient amount of data has been accumulated on the molecular and cellular processes that occur in the early stages, at the trophoblast invasion, the imbalance of which leads to placental insufficiency and, accordingly, the release of various signaling molecules, which can be considered as diagnostic markers long before the onset of clinical symptoms of preeclampsia. In addition, the authors did not define atypical preeclampsia, although this is the subject of a review. Moreover, the predisposing factors and concomitant factors that are associated with the occurrence of atypical PE, in particular, complete or partial hydatidiform mole and triploidy, APS, etc., were not indicated. It would also be appropriate to indicate the use of clinical, biophysical and biochemical markers for screening for early-onset PE in the 1st trimester.

Also, I don't quite understand the following fragment: «Do the processes of placental insufficiency really only occur in the 69second half of pregnancy? If the clinical picture completely coincides with pre-eclampsia 70in the first half of pregnancy, with reasonable suspicion or certainty of chorionic villi damage and features of placental insufficiency should we not recognize that we are dealing 72with pre-eclampsia? This review aimed to investigate whether we should again modify 73its definition.»  It is known that placental dysfunction develops long before the onset of clinical symptoms. In addition, considering the two-wave endovascular trophoblast invasion, disorders can occur both at the beginning of pregnancy and at the end of the first trimester (weeks 14-16). And the increase in blood pressure in the mother during this period may just be a compensatory mechanism for increasing the vascular exchange between the mother and the fetus.

Concerning to the PLGF and sFlt-1. In my opinion, it is too detailed. It is not clear why such a detailed description is given when the results provide limited data on these factors. It can be reduced to key descriptions in the context of preeclampsia. In addition, fragment 163-183 is more appropriate to indicate at the beginning of this section.

Cases presentation. The table does not provide data on the level of platelets, hemoglobin, ALT, AST, LDH, creatinine, protein in the urine, blood pressure. Have they been measured? If yes, please indicate.

Materials and Methods. The authors contribution have to be indicated at the end of the manuscript in the appropriate section.

It is advisable to indicate the inclusion criteria in Fig. 1

Results. General note - the description have to be indicated after the tables and figures.

What does this phrase in Fig. 3 refer to «*Selective termination and live birth of other twin/triplets;» ?

There is no title for table 3. Here – indicate the decoding of HBD and LB.

What do the numbers in the Outcome column mean?

There is no need for Table 4, since you provide references to this articles indicated in the bibliography, and the main reasons for rejection are PE > 20 weeks of gestation. It can be written in one sentence.

Discussion.

«The well-known definition of preeclampsia that we consider in our study seems 286insufficient when confronted with our review results».  I don't quite understand what it seems insufficient. Please explain.

This statement is not entirely justified, considering the missing data. «Women diagnosed with very early onset preeclampsia had severe angiogenesis 293imbalance, detectable by the sFlt-1/PLGF ratio. Data regarding the first-trimester preeclampsia screening was available in only one woman, with a risk of 1:4, but it would 295be reasonable to expect similarly high results in all women endangered with very early 296onset preeclampsia.»  

General comment. Pretty vaguely description. II don't quite understand what data comparison you are describing in the text: received by you with literature search data? In my opinion, if you raise the problem of diagnosing atypical PE, then it is necessary to focus on the mandatory screening of the 1st trimester, which combines biochemical and hemodynamic parameters, as well as predisposing and concomitant factors. At the same time, indicate the specific examination algorithm that you used when examining your clinical cases, which may be an addition to the diagnostic of PE before 20 weeks. 

Author Response

Reviewer 2

Reviewer advice:

Answer:

Abstract. This section should not include titles (1)Introduction, (2)Materials and methods: , (3)Results:. In addition, I did not see an analysis of evidence regarding the role of PLGF and sFlt-1 in the development of the disease in the manuscript, since the authors themselves indicated the absence of data on the levels of placental factors in the presented search cases.

Agree.

Section titles and risk of bias information have been removed.

We presented the role of PLGF and s-Flt-1 in the development of pre-eclampsia in narrative review, not systematic review form. This was the effect of advice from the IJMS Editor to increase attractiveness to the readership.

“In presented cases from authors’ material, all pregnancies ended up with IUFD and SFlt-1/PLGF ratios were significantly elevated in all women”

This section does not fully explain the pathophysiology of preeclampsia, since by now a sufficient amount of data has been accumulated on the molecular and cellular processes that occur in the early stages, at the trophoblast invasion, the imbalance of which leads to placental insufficiency and, accordingly, the release of various signaling molecules, which can be considered as diagnostic markers long before the onset of clinical symptoms of preeclampsia. In addition, the authors did not define atypical preeclampsia, although this is the subject of a review. Moreover, the predisposing factors and concomitant factors that are associated with the occurrence of atypical PE, in particular, complete or partial hydatidiform mole and triploidy, APS, etc., were not indicated. It would also be appropriate to indicate the use of clinical, biophysical and biochemical markers for screening for early-onset PE in the 1st trimester.

We agree that this section is not fully explaining the pathophysiology of preeclampsia. As we know the pathophysiology of preeclampsia is complex, and presenting it in full is far more than we aimed for. We wanted to shortly present the role of the two proteins that are currently used as a marker of preeclampsia in the clinical setting. In our opinion, it would be interesting for clinicians to better understand the role of those proteins.

 As mentioned above, this section was added after the suggestion from IJMS Editor.

«Do the processes of placental insufficiency really only occur in the 69second half of pregnancy? If the clinical picture completely coincides with pre-eclampsia 70in the first half of pregnancy, with reasonable suspicion or certainty of chorionic villi damage and features of placental insufficiency should we not recognize that we are dealing 72with pre-eclampsia? This review aimed to investigate whether we should again modify 73its definition.»

Thank you for this comment.

Corrected to clarify:

“Do the symptoms of placental insufficiency really only occur in the second half of pregnancy? If the clinical picture completely coincides with pre-eclampsia in the first half of pregnancy, with reasonable suspicion or certainty of chorionic villi damage and features of placental insufficiency should we not recognize that we are dealing with is pre-eclampsia?”

Concerning to the PLGF and sFlt-1. In my opinion, it is too detailed. It is not clear why such a detailed description is given when the results provide limited data on these factors. It can be reduced to key descriptions in the context of preeclampsia. In addition, fragment 163-183 is more appropriate to indicate at the beginning of this section.

PLGF and sFlt-1 are widely used in clinical practice as markers of preeclampsia, but usually with little understanding of their role.

We found it extremely interesting to present their role in detail.

Fragment 163-183 was moved to the beginning of the section.

Cases presentation. The table does not provide data on the level of platelets, hemoglobin, ALT, AST, LDH, creatinine, protein in the urine, blood pressure. Have they been measured? If yes, please indicate

Thank you for this comment. Unfortunately, we do not have direct access to the presented cases' medical records as they are in the archives. Obtaining this data is possible, but not in the time indicated for answering reviewers' comments. Therefore we cannot answer clearly if those parameters were checked and their values.

The authors contribution have to be indicated at the end of the manuscript in the appropriate section.

It is advisable to indicate the inclusion criteria in Fig. 1

Agree. The authors' contributions moved.

The inclusion criteria were missing.  Now inclusion criteria are described in fragments 242-244.

What does this phrase in Fig. 3 refer to «*Selective termination and live birth of other twin/triplets;» ?

Thank you for this comment. It refers to the selective termination procedure when in multiple pregnancies one or more fetuses are terminated to achieve the survival of the remaining.

Added “two selective terminations of one fetus with the live birth of another twin/triplet”

There is no title for table 3. Here – indicate the decoding of HBD and LB

Corrected.

Title: Presentation and summary of included cases.

Descriptions of HBD and LB added

What do the numbers in the Outcome column mean?

This are weeks of pregnancy when the pregnancy ended. Included that in decoding of HBD

There is no need for Table 4, since you provide references to this articles indicated in the bibliography, and the main reasons for rejection are PE > 20 weeks of gestation. It can be written in one sentence.

Information provided in Table 4, is mentioned as a requirement in the PRISMA checklist, paragraph 16b. We opt for keeping it in the article.

«The well-known definition of preeclampsia that we consider in our study seems 286insufficient when confronted with our review results».  I don't quite understand what it seems insufficient. Please explain

We meant that the current definition excludes preeclampsia cases before 20 weeks.

We changed the sentence for clarity:

“The International Society of Study of Hypertension in Pregnancy definition of preeclampsia excludes the possibility of preeclampsia before 20 weeks of gestation.”

This statement is not entirely justified, considering the missing data. «Women diagnosed with very early onset preeclampsia had severe angiogenesis 293imbalance, detectable by the sFlt-1/PLGF ratio. Data regarding the first-trimester preeclampsia screening was available in only one woman, with a risk of 1:4, but it would 295be reasonable to expect similarly high results in all women endangered with very early 296onset preeclampsia.»  

Agree. We meant patients presented in our cases. Added this information for clarity.

In our presented cases, women diagnosed with very early onset preeclampsia had severe angiogenesis imbalance, detectable by the sFlt-1/PLGF ratio”

General comment. Pretty vaguely description. II don't quite understand what data comparison you are describing in the text: received by you with literature search data? In my opinion, if you raise the problem of diagnosing atypical PE, then it is necessary to focus on the mandatory screening of the 1st trimester, which combines biochemical and hemodynamic parameters, as well as predisposing and concomitant factors. At the same time, indicate the specific examination algorithm that you used when examining your clinical cases, which may be an addition to the diagnostic of PE before 20 weeks.

Thank you for this comment.

 We added information that all authors’ cases were screened according to FMF protocol.

We have also added a paragraph to the discussion regarding first-trimester screening.

In light of the available evidence, it is not possible to assess the performance of first-trimester screening in the prediction of atypical preeclampsia. On the other hand, all our presented cases had high highly altered PLGF/sFlt-1 index at the onset of the disease. PLGF is recommended in preeclampsia screening according to the Fetal Medicine Foundation, and therefore we think that screening including all recommended maternal characteristics, blood pressure, uterine arteries flow, and biochemical markers including the PLGF may be enough to screen for atypical preeclampsia.”

Round 2

Reviewer 1 Report

I agree with answer for suggestion number 1. I do think  its worth publishing in this form

Reviewer 2 Report

I have no comments